# Donor activity is associated with US legislators' attention to political issues

**Pranav Goel** [1]*, **Nikolay Malkin**[2,3], **SoRelle W. Gaynor**[4], **Nebojsa Jojic**[5], **Kristina Miler**[6], **Philip Resnik**[7,8]

**1** Department of Computer Science, University of Maryland, College Park, Maryland, United States of America, **2** Mila, Québec AI Institute, Montréal, Québec, Canada, **3** Department of Informatics and Operations Research, Université de Montréal, Montréal, Québec, Canada, **4** Department of Political Science, College of the Holy Cross, Worcester, Massachusetts, United States of America, **5** Microsoft Research, Redmond, Washington, United States of America, **6** Department of Government and Politics, University of Maryland, College Park, Maryland, United States of America, **7** Department of Linguistics, University of Maryland, College Park, Maryland, United States of America, **8** Institute for Advanced Computer Studies, University of Maryland, College Park, Maryland, United States of America

⊛ These authors contributed equally to this work.
* pgoel1@cs.umd.edu

**Data Availability Statement:** Our data files and replication code including expert annotations are all publicly available as a compiled repository and can be downloaded at: https://zenodo.org/record/7465346 (DOI: 10.5281/zenodo.7465346).

## Abstract

Campaign contributions are a staple of congressional life. Yet, the search for tangible effects of congressional donations often focuses on the association between contributions and votes on congressional bills. We present an alternative approach by considering the relationship between money and legislators' speech. Floor speeches are an important component of congressional behavior, and reflect a legislator's policy priorities and positions in a way that voting cannot. Our research provides the first comprehensive analysis of the association between a legislator's campaign donors and the policy issues they prioritize with congressional speech. Ultimately, we find a robust relationship between donors and speech, indicating a more pervasive role of money in politics than previously assumed. We use a machine learning framework on a new dataset that brings together legislator metadata for all representatives in the US House between 1995 and 2018, including committee assignments, legislative speech, donation records, and information about Political Action Committees. We compare information about donations against other potential explanatory variables, such as party affiliation, home state, and committee assignments, and find that donors consistently have the strongest association with legislators' issue-attention. We further contribute a procedure for identifying speech and donation events that occur in close proximity to one another and share meaningful connections, identifying the proverbial needles in the haystack of speech and donation activity in Congress which may be cases of interest for investigative journalism. Taken together, our framework, data, and findings can help increase the transparency of the role of money in politics.

## Introduction

A majority of publicly-traded US-based corporations as well as many labor unions and interest groups organize a Political Action Committee (PAC) to raise funds and donate money to

**Funding:** National Science Foundation grant 2008761 (PG, SWG, KM, PR) (https://www.nsf.gov/awardsearch/showAward?AWD_ID=2008761) National Science Foundation grant 2031736 (PG, PR) (https://nsf.gov/awardsearch/showAward?AWD_ID=2031736) Amazon (PG, PR) (https://www.amazon.science/research-awards/program-updates/2020-amazon-research-awards-recipients-announced) The funders had no role in study design, data collection and analysis, decision to publish, or preparation of the manuscript.

**Competing interests:** The authors have declared that no competing interests exist.

candidates running for political offices. PACs are major players in American politics, contributing nearly $500 million to congressional candidates in the 2020 election cycle alone (source: https://www.opensecrets.org/political-action-committees-pacs/2022). They play a significant role in funding congressional campaigns [1–3] and have been found to exert influence on the outcome of elections [2, 4, 5]. Since legislators rely heavily on donations to meet the high and rising costs of campaigns, as well as to demonstrate their electoral strength [6], soliciting donations from PACs is an important and time-consuming element of legislators' day-to-day work [7] (see also: https://www.cbsnews.com/news/60-minutes-are-members-of-congress-becoming-telemarketers/). There is a rich literature on PACs' decision to donate and the impact of those donations on legislators and policy outcomes. A large body of work considers the relationship between PACs and congressional elections, including the effect of incumbency advantage, differences in media costs, and voter attention [8–11]. Within the institution, pPrior work has considered how PACs strategically target legislators based on their existing policy positions and membership on relevant congressional committees [12–16]. Other research finds evidence that donors contribute to legislators who share ideological preferences [17–19], and that donors are more likely to gain access to legislators [4, 16, 20–27].

Research on donor activity and legislators' votes on congressional bills, however, comes to mixed conclusions regarding correlations and even less support for a causal relationship [12, 28–31]. While there are some cases of PAC influence on specific roll-call votes [32, 33], there are also numerous examples of a lack of relationship between money and votes [34, 35]. The existing work likely reflects the complex reality that donations can be both an incentive and a reward, and are frequently made to legislators who tend to agree with the PAC's preferences, all of which makes isolating the causal impact of money difficult. Even without definitive findings, this body of research raises concerns about the potential role of money in politics, particularly given the bias towards economic organized interests [36, 37].

Identifying reliable associations is the starting point to understand potentially important relationships between donations and congressional behavior. Towards this goal, scholars have turned to examining the impact of money on behaviors beyond votes. While roll call votes constitute an important legislative action, they are limited in their ability to capture the full range of positions and priorities of individual legislators, and thus perhaps less affected by donors. Rank-and-file members have little autonomy over the congressional agenda, and the binary (yes-or-no) nature of roll call votes limits the positions that votes can convey. In an increasingly centralized and partisan legislative environment, members are often forced to "toe the party line" and lose the chance to carve a separate position for themselves either within their party or in the congressional chamber as a whole [38–40]. Conversely, legislators have much more discretion over their participation in other parts of the legislative process as well as their *rhetoric* [41–43]. For instance, scholars have found evidence that donors affect legislators' behavior in committees [12, 15, 35, 44]. More recent work has begun to investigate the relationship of donors to legislative rhetoric, including how donors influence the adoption of suggested legislative text [45, 46], the strategic use of interest groups' committee speech to influence legislators' policy positions [47], and the bidirectional relationship between donors and US senators when examining committee discussions on energy policy [48].

Our work contributes a new avenue to the study of the association between money and non-voting behavior by focusing on the issue agenda in Congress and the specific issues that legislators choose to call attention to on the House floor. Floor speeches allow individual legislators to convey proximity to party leadership or distance themselves from the party "message" on any policy [39]. And while there are some limitations when it comes to scheduling and time allotted to speak (determined by the House Committee on Ethics), and the restriction of members not being allowed to explicitly campaign on the House floor or use floor speeches

directly in campaign materials, floor speeches provide a rich view of a legislator's thinking and priorities [49, 50] via the attention allocated to various policy issues (*issue-attention*) [51] and framing [52].

Legislators' issue-attention is tied to the agenda-setting function of communication and rhetoric by the political elite. Prior work has considered issue-attention *i)* a direct proxy of issue prioritization by policymakers [53], *ii)* an important criterion for driving policy action both inside Congress [54, 55] as well as outside Congress [56], *iii)* an influence on public opinion [57], and *iv)* a powerful aspect of wielding political power, especially when attention influences which issues become or do not become a part of the policy agenda in Congress [58]. This can be of great importance to underrepresented groups in Congress [51, 59, 60]. Floor speeches are an avenue for members of Congress to proactively show their expertise on policy issues [61], increase their visibility, and demonstrate their commitment to particular issues and stances—not just to their fellow representatives but to the press and their constituents [62–64] as well as outside political actors including PACs. With reelection-minded legislators looking to communicate their preferences to their constituents [65], floor speeches are an important tool for legislators to convey how they represent their constituents and to create a particular public image [42] that is often not possible through roll-call votes. Legislators can also use speeches to signal to PACs and interest groups their shared preferences and willingness to advocate on those issues. Our work can therefore also help document the professional relationships that PACs and legislators seek to build [24, 66].

The general importance of PAC donations to a legislator's political career, as well as the importance of floor speeches as a critical legislative action that offers insights into congressional agenda setting and policy attention, motivates us to conduct the first examination of the association between donations made by PACs and the attention that legislators place on various issues in their floor speeches. Although this research remains agnostic regarding causal directions, we uncover a strong and robust association between money and legislative speech, which reveals that money has a more extensive reach in American politics than previously established and proposes a foundation for future work to investigate the potential causal relationship.

In this work, we seek to answer the following question: **are donors substantially associated with legislators' issue-attention when speaking on the floor of Congress?** To study this, we compare the association provided by information about donors with other information about legislators such as their state, party affiliation, and committee assignments (which serve as alternative explanations of legislative decision-making). We find that donor information consistently offers the strongest association with legislators' issue-attention. This uncovers a previously unidentified relationship and provides new evidence of how money and issue priorities go hand-in-hand. This association can help inform our understanding of legislative motivations and enable greater transparency and accountability of elected officials.

In addition to our main finding that donor information is meaningfully associated with legislators' issue-attention in their floor speeches (both in the aggregate and across individual congressional sessions), we gain new insights into PAC dynamics by uncovering both the top donors within each issue area, and the top issues for each industry.

We create a new dataset that includes information and metadata about legislators, along with the text and dates of floor speeches, donations they receive from PACs, and information about the PACs such as the industry they belong to. Our comprehensive dataset covers the US House of Representatives over twelve congressional cycles, from 1995 to 2018 (which encompasses periods of both Republican and Democratic control). We note that the US Supreme Court's ruling in *Citizens United v. FEC* (2010) affected campaign finance law in the area of outside independent spending in federal campaigns. The ruling gave rise to independent

expenditure-only PACs (also called Super PACs), but these are distinct from the type of traditional hard-money PACs examined here. Traditional PACs donate money directly to candidates' campaigns and were not the focus of the Citizens United case.

Using our new dataset, model, and the expert validation of issue-donor associations given by the model, we also introduce a procedure that highlights specific donation and speech events that *a)* occur in close proximity to each other, and *b)* stand out as significantly connected (*i.e.*, are more likely to *not* constitute a random occurrence of a pair of events close in time). This procedure can find cases where a donor donates a significantly large amount to a legislator within days (before or after) of the legislator giving a floor speech on a policy issue that the donor cares about (an example is provided in the next section). We identify these cases considering every congressional cycle in isolation, and can score and rank them based on the strength of the association (detailed procedure provided in S10 Appendix). Experts and journalists can then use the filtered set of connected donation and speech events occurring within a small time window (less than a month; before or after) as starting points to identify situations that potentially warrant further investigation. Our codebase documents this procedure and the ranked cases. Our dataset, model, and analyses will enable further work on the connections between legislators' language use and their donors.

## Materials and methods

### Data

Our dataset spans the US House of Representatives from 1995–2018. The US Congressional Record database provides transcripts of floor speeches as well as various information and metadata about legislators [67] (see: https://github.com/unitedstates/congressional-record for floor speeches transcripts, and—https://github.com/unitedstates/congress-legislators—for biographical information). We also use a publicly available resource on compiled congressional committee assignments [68]. This committee assignment data includes select as well as joint committees and the various modified committees created over the years covered in our dataset. Raw data on US political campaign donations is made available by public governmental databases like the Federal Elections Commission (FEC) (https://www.fec.gov/); we use publicly available bulk data on donation transactions as well as donor s provided by OpenSecrets, a nonprofit research organization (https://www.opensecrets.org/bulk-data/). We detail the process of linking together these databases to create our new dataset to enable our research and an overview of the contents of our database including details on the annotations for industrial groupings for PACs (Industry, Category) in S1 Appendix.

In our processed dataset (consisting of business and labor PACs), most PACs donate to fewer than a third of legislators and therefore are somewhat targeted in their donations (S2 Fig). In addition, most PACs in our data are not donating in a clearly partisan fashion; instead, they donate to a mix of Democratic and Republican legislators (S3 Fig), as prior work has found is often the case for policy-oriented or connected PACs. Our processed dataset consists of both business (or corporate) and labor PACs. We detail statistics about the relative donation patterns of these two types of PACs and compare their association with legislators' issue-attention in S14 Appendix and S22 Fig. Business PACs donate to more legislators on average in our data, and also have a significantly higher association with issue-attention than labor PACs (S22 Fig). This novel examination adds to prior work that also focuses on the critical dynamics of the influence of business and labor money in American politics [45, 51, 69–75]. We discuss our dataset schematic (S1 Fig) in S1 Appendix, and provide details about data processing along with final data statistics in S2 Appendix.

Our data files and replication code including expert annotations are all publicly available as a compiled repository and can be downloaded at: https://zenodo.org/record/7465346 (DOI: 10.5281/zenodo.7465346) [76].

## Method

An illustrative example using our actual data and method is displayed in Fig 1B to provide a visual overview of our methodology as well as the intuition behind our framework.

**Topic modeling.** Issue-attention is quantified using a topic model [77], which uses the content of the floor speeches to identify a collection of topics or themes; each speech given by a legislator is then considered a mix of the discovered topics. Using discovered issues (topics) from text as a key component in our analysis follows previous scholarship in political science [43, 49]. In fact, pertaining to campaign contributions, a particular prior work studied PAC influence on roll call votes and committee deliberation in the US Congress and found that the manner of the PAC influence is linked to the type of issue and the issue context [78]. Corporate funding has been found to influence the themes present in discourse around climate change (specifically, discourse going against the scientific consensus is impacted by funding from corporations and organizations dedicated to this alternative climate change narrative) [79].

Specifically, in this work, we use LDA estimated with Gibbs Sampling [77, 80] as implemented in the MALLET package [81] as our choice of topic modeling method. LDA models a collection of documents through a collection of latent topics or themes. Each topic is a probability distribution over the words in the vocabulary, and each document is represented as a mixture of topics or themes. The corresponding value for a given topic in the probability distribution over topics for a speech quantifies the presence of that topic in that particular speech.

Prior work has found classical LDA to be the dominant topic model of choice among practitioners and that it yields strong qualitative ratings for its topics as judged by humans [82]. Recent advances in the machine learning and natural language processing literature have introduced new types of topic models, with one such noteworthy introduction being neural topic models that utilize deep neural networks [83] (see [84] for a recent survey on neural topic modeling). However, these new topic models had claimed improvement over LDA via an automatically computed measurement that has since been established as an *invalid* proxy of human judgment [82]. Further, more recent work shows that LDA is a more reliable choice for qualitative content analysis than recent neural topic modeling methods [85]. Taken together, these two works make a strong case for LDA as the choice of topic modeling method for qualitative content analysis in order to interpret latent categorical themes present in a document collection (containing English text). We use LDA motivated by this body of recent work in topic modeling, but in order to establish the robustness of our main findings with respect to the choice of topic modeling method, we also present our main result with three different topic modeling methods: the Structured Topic Model or STM [86, 87]; Non-negative Matrix Factorization or NMF [88–90]; and the Contextualized Topic Model or CTM, which is a neural topic modeling method that uses contextualized word embeddings [91] (S14–S16 Figs; details are provided in S11 Appendix).

Topic modeling is an unsupervised method. In order to enhance the interpretability and quality of this computational model's output, meaningful topics are identified and explicitly labeled with the policy issue they represent by two experts (political scientists; examples shown in Fig 1A), who use the estimates provided by the topic model (each topic is viewed in terms of the top representative terms and floor speeches in the data by the political scientists). The two experts independently labeled each topic in our work. The complete set of instructions used for obtaining labels for topics and annotations are provided along with our data. Some

| Topic (top terms) | Label (Expert 1) | Label (Expert 2) |
|---|---|---|
| education, schools, students, teachers, districts, parents, kids, teacher, quality, educational | Education (K-12) | Education |
| medical, patients, doctors, hospital, patient, hospitals, doctor, quality, providers, physicians | Healthcare | Health care |
| drug, drugs, tobacco, abuse, colombia, fda, substance, illegal, marijuana, addiction | Drugs | Drugs |
| veterans, va, benefits, affairs, deserve, receive, disabled, disability, disabilities, families | Veterans | Veterans |
| education, college, students, programs, training, student, workforce, institutions, opportunities, loans | Education (higher ed) | Education |

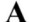

**A**

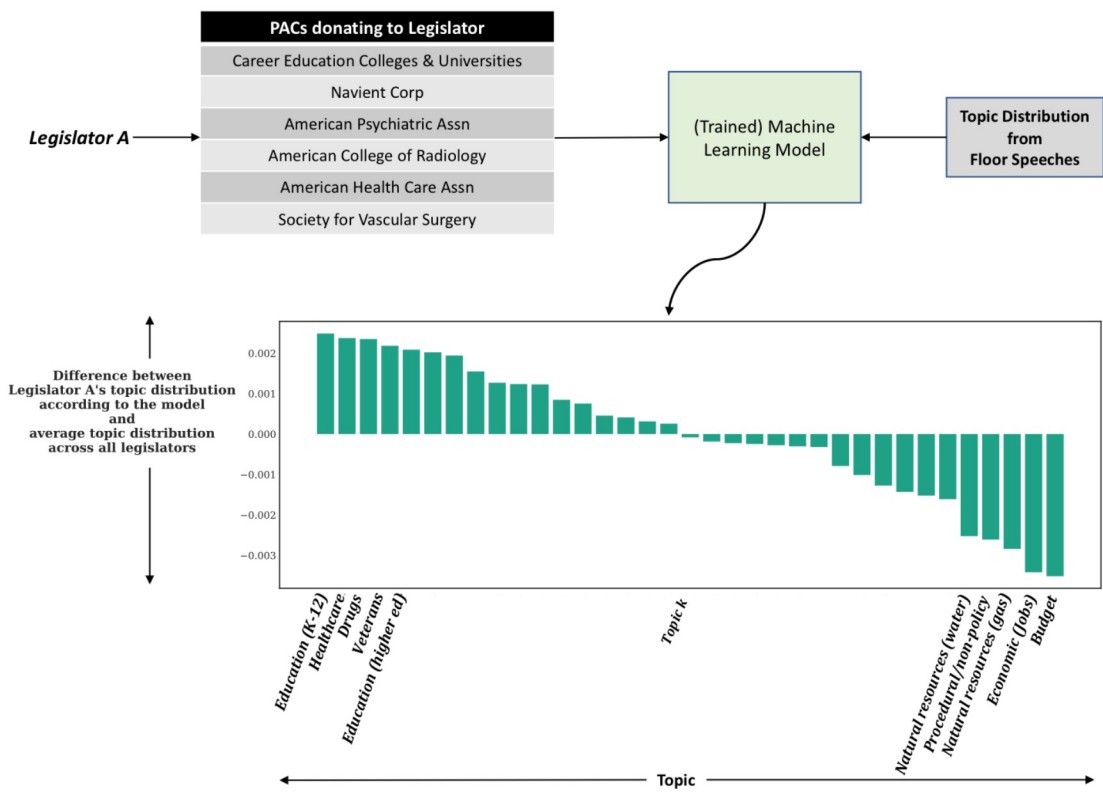

**B**

**Fig 1. Illustration of our unsupervised machine learning approach of investigating issue-attention in floor speeches given by individual legislators and the association with their donors.** First, we infer topics from the collection of all floor speeches given by legislators in our processed dataset. **A**: illustrates the inferred topics by showing the top terms; we asked experts to label the topics (based on the top terms as well as the top documents). Each legislator is then represented by a topic distribution for their speeches in a session or over the entire period in our dataset as well as the distribution over donors (PACs) from whom the legislator received support. Then, we use all legislators in a training set to train a machine learning model that predicts the topic distribution for legislator using their PAC distribution as features. We can then analyze the difference between the predicted distribution and the average topic distribution over all legislators to detect links between PACs and floor speeches. **B**: shows one such distributional difference for a simulated individual 'Legislator A', with issues with the most disproportionate focus shown on the left and least such focus shown on the right. With healthcare and education-related PACs donating to this legislator, the model trained to capture associations between donors and issue-attention expects a disproportionate focus on education and healthcare-related issues compared with average attention paid to those issues across all legislators and their speeches in the data. The x axis uses the issue labels provided by one of our experts (expert 1) for the topics. "Navient Corporation is an American student loan servicer based in Wilmington, Delaware" (source: https://en.wikipedia.org/wiki/Navient).

examples of topics and labels are shown in Fig 1A, and additional examples are displayed in S3 Table. We note that while experts tend to come up with the same policy or issue labels for topics, sometimes there are differences in the level of specificity between the two experts—for example, 'education' versus 'higher education'.

We provide results after non-meaningful, non-policy-related topics, per the judgment of either of the two experts, are removed—we explicate which labels were removed in S4 Appendix, and we observe no change in our findings and interpretation of the results (S5 Appendix, S4 Fig). Therefore, the findings in this work are not dependent on automatically discovered topics alone, but in fact, are reliably established across both the initial set of automatically discovered topics as well as the expert-refined or curated set of issues.

**Note on implementation details.** We use Gensim's Python wrapper for MALLET LDA: radimrehurek.com/gensim_3.8.3/models/wrappers/ldamallet.html. We establish that our main result is not dependent on the order of the documents used to train the LDA topic model (S12 Appendix).

**Machine learning based experimental framework.** One of the outputs of the topic model's training is the posterior distribution over the discovered topics for each floor speech. This is a representation of each speech as a collection of $K$ positive weights associated with the topics. For each legislator in the processed data, we average their topic distributions over all the speeches they gave on the House floor to get one topic distribution that indicates how much the legislator spoke on each topic on average across all the speeches they gave—we refer to this particular topic distribution as legislators' issue-attention.

In this work, we refer to information about legislators as a legislator attribute. We include non-donor-related legislator attributes such as party affiliation (*Party*), home state (*State*), and committee assignments (*Committee*). For donor-related legislator attributes, we consider which individual PACs donated to the legislator (*PAC*) in order to assess the donor-legislator association at the heart of this study. We also consider the granularity of the information on donors by comparing donations from individual PACs against donations made by a specific industry (e.g., 'Public Unions') or a particular subset of the industry (e.g., 'Teacher Unions')—giving us two more donor-based attributes in addition to the individual donor: PAC *Industry* and PAC *Category*.

Finally, we consider another baseline legislator attribute for comparison called *Random-PAC*, where the *PAC* attribute is 'shuffled' so that each legislator is represented using the donor (PAC) profile of a random, different legislator. Since various legislator attributes have different numbers of possible values (for example, three for *Party*: Democrat, Republican, and Independent), and the number of PACs in our data is orders of magnitude larger than other attributes, this serves as a check for the effect of legislator attribute size. Specifically, we have 1002 individual PACs in our final processed dataset, while other attributes have fewer possible values (or sizes), such as 3 for *Party*; details on the size of each attribute-set are included in S1 and S2 Tables. In addition, this serves as a robustness check for our findings (as a null test): if *Random-PAC* is associated with issue-attention as well as *PAC*, that would indicate that knowing which particular PACs donated to a legislator under consideration does not have any association with their issue-attention (allowing us to test a statistically meaningful signal versus chance).

We note that all the above attributes are represented as one-hot encodings (or binary vectors where each dimension corresponds to presence or absence of a possible value of the attribute). We investigate the impact of these attributes by measuring the strength of their association with legislators' issue-attention in their speeches, drawing on a specific machine learning approach. In machine learning frameworks, the strength of association between attributes and a dependent variable (in our case, issue-attention) is measured in terms of the *predictive capacity or strength* of various attributes.

Specifically, let the matrix of averaged topic probabilities (number of legislators × number of topics) be *y* (dependent variable) and a given legislator attribute be *X* (independent variable). To predict *y* from *X*, we train a multinomial logistic regression model in a machine

learning framework:

$$y' = P(X) = \frac{1}{1 + \exp^{-(\beta_0 + \beta_1 X)}} \tag{1}$$

$\beta_0$ is the bias or intercept term, and $\beta_1$ is the weight term (together, these are the regression coefficients of our model). $y'$ is the predicted topic probability distribution for the $K$ topics, and it can be compared with the actual topic distribution ($y$) derived from floor speeches in order to train the model. If the dimensionality (or size) of the independent variable $X$ is $m$, then the weight matrix $\beta_1$ will be of the size $K$ by $m$ (for e.g., 60×1002 in case of the legislator attribute being *PAC*). This weight term of the trained multinomial logistic regression model indicates the learned associations between each of the values of an attribute (such as an individual PAC) and each of the topics. We use $L_2$ regularization for our multinomial logistic regression model (training details are provided in S6 Appendix).

In our framework, the multinomial regularized logistic regression model is trained to predict a probability distribution vector with $K$ values from a legislator attribute or representation. To assess the predictive strength of a particular legislator attribute, we use the standard machine learning methodology of cross-validation—legislators are divided into a 'training set' (used by the model to learn how to map legislators represented using an attribute to their provided issue-attention, derived from floor speeches) and a 'held-out set', where the trained machine learning model uses the learned mapping to 'predict' legislators' issue-attention using the attribute information. On the held-out set of legislators, predicted issue-attention is compared with actual issue-attention for those legislators. In a cross-validation setting, this procedure of dividing legislators into two sets is repeated multiple times, and results can be shown and compared using an average across held-out sets.

When examining the predictive capacity of various legislator attributes for issue-attention over the entire set of floor speeches from 1995–2018 considered together, we run LDA with sixty topics ($K = 60$). We note again that we do not only rely on the obtained topic distribution for our main finding, but conduct an extensive expert curation process to additionally obtain a coherent set of labels for political issues used in our analyses (step 1 in the human curation procedure discussed in S4 Appendix). We show that after removing non-coherent or non-substantive categories or issues (that reduces the initial number of topics (60) to a set of identified issue categories based on expert judgment (48)), our main finding continues to hold (S5 Appendix and S4 Fig). We also establish that our main result holds across different choices of the number of topics, including $K = 30$, $K = 45$, $K = 90$, and $K = 120$, in S13 Appendix.

We also train a separate 30-topic model on floor speeches made within each of the twelve congressional cycles (1995–96 to 2017–18) in our data (we lower the $K$ to account for the lower number of speeches). This helps us focus on a two-year congressional session at a time and analyze how the results change over time.

**Measuring the association.**  To measure the association between a legislator attribute and issue-attention, we recognize that the machine learning setup converts legislator attributes into a 'predicted' issue-attention distribution, which can be compared with the actual issue-attention distribution derived from floor speeches. Since we are dealing with probability distributions, we use the information-theoretic concept of optimal encoding of distributions for our measurement.

Consider two sets of legislators: one s in the training set, and one s in the held-out set. The goal for our machine learning method trained using a particular legislator attribute (as the 'model') is to predict or provide its best estimate of issue-attention on the held-out set. The expected lowest number of bits required to encode issue-attention for the held-out set is the

negative entropy of the distribution. This provides us with the *lower-bound*. Note that a lower number of bits is better since it implies a more efficient (or optimal) encoding of a distribution.

In the absence of any information on legislators, we can use the issue-attention available in the training set, and measure the cross-entropy between training set and held-out set issue-attention (topic distributions) in bits of information. This provides us with an *upper-bound*. In the absence of information on legislators (any of our legislator attributes), the maximum distance (in terms of bits of information) we could reduce is given by the difference between this upper bound and the aforementioned lower-bound.

In the *presence* of information about legislators, such as which PACs donated to them, we can use our machine learning method to obtain an estimate for the held-out issue-attention. Cross-entropy once again measures the bits required to encode the actual held-out issue-attention given the estimated issue-attention provided by a model of the legislator. We call this the *model cross-entropy*. Our measurement (used in the Results section below) is called the **% bit reduction**, and is computed as follows:

$$\%\text{Bit Reduction} = 100 * \frac{\text{upper} - \text{bound} - \text{model cross} - \text{entropy}}{\text{upper} - \text{bound} - \text{lower} - \text{bound}} \qquad (2)$$

Ultimately, our measurement computes the reduction in bits required for encoding issue-attention—a higher reduction implies that a more optimal encoding is enabled by relevant information about legislators (which can also be framed as the particular legislator attribute offers more association with the variation in the issue-attention of the legislators). This measurement helps us both compare various legislator attributes, and quantify the explanation offered by the attribute (as a %).

## Results

### PACs are significantly more associated with legislators' issue-attention in floor speeches than other legislator attributes

In this set of results and discussion, we consider information about legislators and the content of all their floor speeches over the entire period of our data (1995–2018).

Fig 2A shows that knowing which PACs donated to a legislator explains more of the variation in legislators' issue-attention than knowledge of other legislator attributes. This effect is statistically significant ($N = 50$, $p < 0.05$; details provided in S7 Appendix) and reveals that *PAC* information provides the strongest association with legislators' issue-attention. Other non-donation-related attributes about legislators, such as their party, home state, and committee assignments are significantly less predictive of the issues that legislators talk about than PAC donations. This is a striking finding because one expects parties, committees, and state interests to be tightly related to the issues that legislators spend their time talking about in the House. Notably, information about individual PACs (such as 'Career Education College & Universities' PAC) is more predictive than information about industry-based groups of PACs (such as the *Industry*: 'Education'; or even the specific *Category* within the industry: 'For-profit education').

Finally, we find that the effect is not simply due to the higher number of possible values (or dimensionality) for the *PAC* attribute as shown by the baseline attribute of *Random-PAC*, where the donor profile of legislators is randomly shuffled. We note that the negative result for *Random-PAC* is due to the machine learning model *overfitting* to the training data. In other words, the information did not allow the model to generalize in its predictions of issue-attention for legislators not seen during the training of the model (the held-out set, as discussed

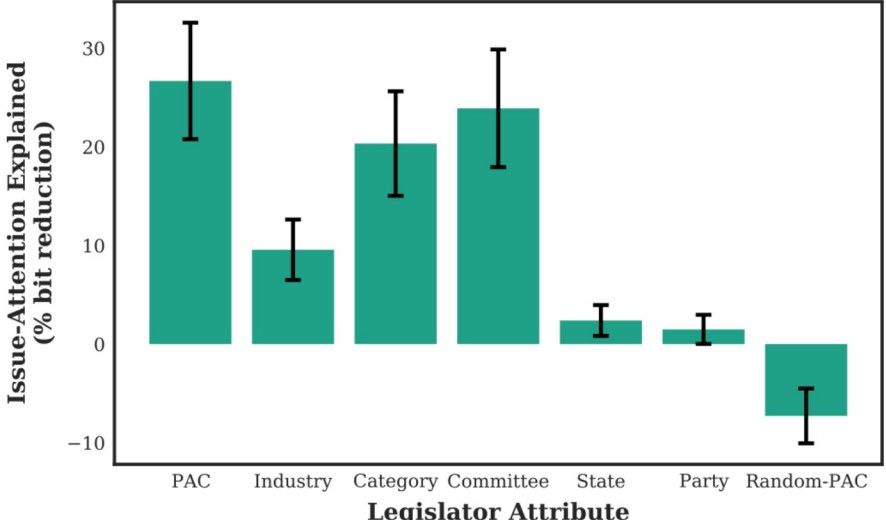

**A**

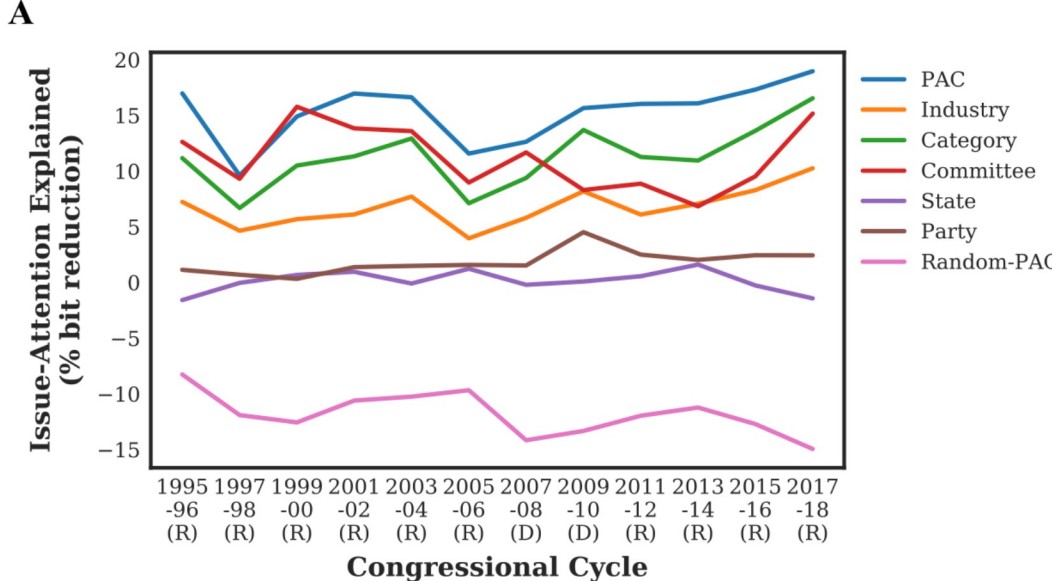

**B**

**Fig 2. PACs are most associated with legislators' issue-attention in floor speeches compared with other legislator attributes, both in aggregate terms across the entire time period of our dataset as well as in individual congressional cycles.** We show how much various legislator attributes explain legislator's issue-attention and how they compare with one another. Issue-attention is instantiated using topic distributions derived from floor speeches. A machine learning framework learns associations between legislator attribute values and their corresponding topic distribution on a training set, and then predicts the topic distributions for legislators on a held-out or validation set (*i.e.*, these legislators are not used in the training of the model). **A**: shows the results in aggregate, with information about which individual PACs donated to legislators clearly explaining their issue-attention the most out of all the legislator attributes we consider; the cross-validation validation procedure was repeated 50 times, and the error bars (for results across the cross-validation procedure) are shown. **B**: shows the results when repeating our topic modeling and machine learning procedure separately for each of the twelve congressional cycles in our data, and demonstrates the same pattern as the aggregate picture (with the cross-validation procedure repeated 30 times; the error bars are provided in S5 Fig). (R) and (D) indicate which party was in the majority in the US House during that congressional cycle. Both of our findings are statistically significant (S7 Appendix).

above in the method overview). This 'overfitting' effect for *Random-PAC* is made clear by viewing the results on the training set provided in S6 and S7 Figs.

## PACs consistently offer the strongest association with issue-attention across all congressional cycles

In addition to training and using one topic model and machine learning model for the entire data spanning twelve congressional cycles, we also train separate topic models and machine learning models for each congressional cycle following the same process (with only a few training details and choices changed as detailed in S6 Appendix). The same legislator attributes are compared in terms of their predictive power for House representatives' issue-attention in each two-year congressional cycle. We find that *PACs* are the best predictive indicator of legislative issue-attention for every cycle (Fig 2B). The difference between the predictive strength of *PAC* compared to other attributes is significant for most cycles ($N = 30$, $p < 0.05$, details in S7 Appendix).

Apart from donation information, the only other competitive legislator attributes are *Committee* assignments. We explore the comparison for *PAC* and *Committee* in greater detail in S8 Appendix, and also find that combining these two attributes does better than using just one or the other (which is not the case for combining other attributes with *PAC* (S9 Appendix)). This indicates that there is at least some complementary predictive power present in these two sources of information on the legislators (S8 and S9 Figs).

Conducting our experiment separately for each congressional session also allows us to quantify two other legislator attributes: legislator *Seniority*, based on the number of congressional terms the legislator has served up to and including the particular congressional session being analyzed; and legislators' *District Marginality*, based on how close their district was in the most recent US presidential election in its vote for the Democratic and Republican party presidential candidates. Construction of these two additional attributes and their results are discussed in S15 Appendix (with results visually presented in S23 and S24 Figs). The results conform to the aforementioned finding; in fact, seniority and marginal district values are unable to explain issue-attention with any significance.

## Meaningful issue-PAC associations can be automatically uncovered

As part of learning the association between a legislator attribute and issue-attention (topic distribution), our machine learning model uses the data to quantify associations between each individual attribute value (such as each individual PAC) and attention paid to each individual issue. We use these automatically discovered quantitative values signifying issue-PAC associations to assess if they are meaningful to political scientists (experts)—this helps validate that the results found in the aggregate (in terms of explaining issue-attention) are derived from non-arbitrary human-understandable relationships.

Using these values for issue-PAC associations as identified by our models when trained on the entire data, we get the top 10 PACs (representing $\sim 1\%$ of all PACs in the processed data) for every issue. Note that we only consider meaningful topics here as issues per the labels created by political scientists (the discarded labels are provided in S4 Appendix). Two political scientists rated each PAC shown for an issue as *Related* (3), *Potentially related* (2), or *Unrelated* (1). In addition to the name of the PAC, experts are also provided information about the industrial sector of the PAC and are free to search for additional information in order to make their judgments. In instances where the two experts' ratings can be compared, we find moderate agreement for issue-PAC associations (details of the inter-expert agreement are provided in S4 Appendix). Examples of issue labels created by experts for a topic and the top 10 PACs

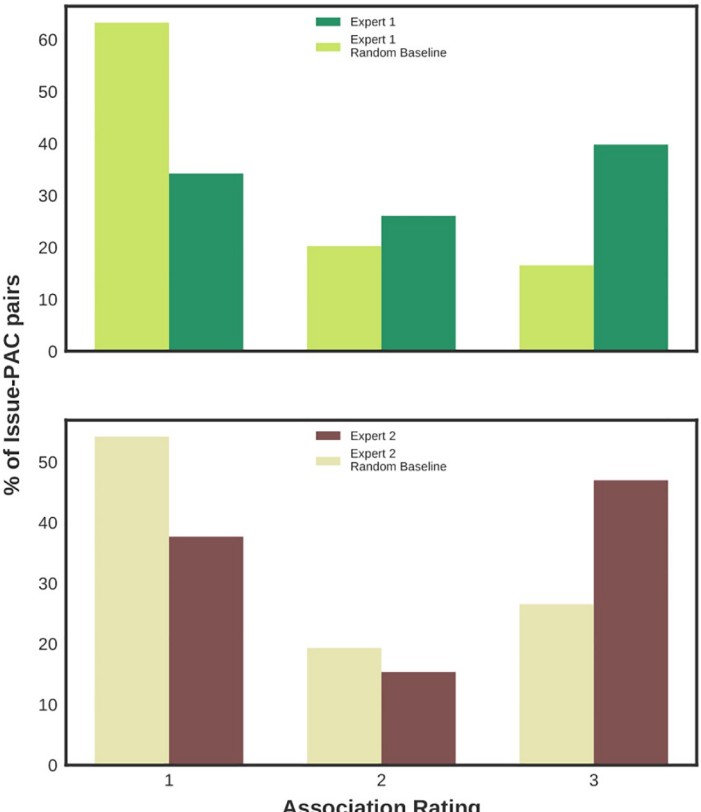

**Fig 3. Issue-PAC associations discovered by the model are deemed meaningful by experts.** Two political scientists independently rated PACs shown for an issue on a 3-point Likert scale (3: the PAC and the issue are related, 1: they are unrelated). The top PACs for each issue per our model are rated higher than a random selection of PACs by both of our experts. We use these manual issue-PAC association or relatedness scores in further analyses.

for the issue per the model are provided in S3 Table. Additionally, to assess whether the ratings for top PACs for issues as given by our model are more meaningful than we would otherwise expect, the same ratings task was repeated with the same setup except the 10 PACs shown for every issue were selected randomly from all the PACs in our data. The experts were not privy to this difference between the two setups—they were given the two tasks with the same instructions. We refer to this latter setup as the 'Random Baseline'.

Fig 3 shows that the PACs found to be most associated with issues by our model are deemed as related by experts at a higher rate than for randomly selected PACs for those same issues; we find that this difference is statistically significant for both experts ($N = 400$, $p < 0.00001$; details in S4 Appendix). This shows that our approach enables a large-scale probabilistic analysis that uncovers real, expert-validated signal in the big donations and unstructured language data. The details of the expert judgment or ratings task including examples of how the annotation task is presented to experts are provided in S4 Appendix, S3 Table, and S11 Fig. The task instructions and final annotations are included in our data and code repository [76].

## Meaningfully connected speech and donation events occurring in close proximity to one another can be surfaced for further investigation

Finally, we contribute a procedure that can identify related donation and speech events in any given congressional cycle in our data that occur within small time windows (such as seven,

**Table 1. An example of the real-world relationships our model and analysis can automatically capture.** On Feb. 25, 2015 and Feb. 26, 2015, Rep. John Kline (R-MN) made several speeches arguing against federal funding for education and in support of charter schools. The PAC Career Education Colleges and Universities ("a Washington, D.C.-based trade association that represents for-profit colleges." per Wikipedia: https://en.wikipedia.org/wiki/Career_Education_Colleges_and_Universities) made a donation of $5,000 to Kline on Feb. 27, 2015—well above the PAC's average (standard deviation) donation amount to any legislator during the cycle, $1,862.16 (±$1,288.27). While these relationships can be uncovered by journalists (see the last three columns of the table), this can be a time-intensive process. Our data modeling uncovered this speech-PAC relationship without supervision.

| PAC name | Topic label (Expert 1/ Expert 2) | Legislator (Affiliation) | Article URL (for face validity) | Article headline (for face validity) | Relevant text snippet from the article (for face validity) |
|---|---|---|---|---|---|
| Career Education Colleges and Universities | Education (higher ed)/ Education | John Kline (R-MN) | www.usatoday.com/story/news/politics/2013/07/23/for-profit-colleges-contributions-house-regulations/2579041/ | For-profit colleges giving big to helpful House members | "House Education Committee Chairman Rep. John Kline, who saw a dramatic upsurge in campaign contributions from for-profit colleges in recent months, is pushing legislation that would help the industry preserve its access to federal student loans." |

fifteen, or thirty days before or after), where the temporal co-occurrence is more likely *not* to be statistically random. We are able to identify cases where a PAC is *significantly more likely to increase their donation activity close to floor speeches made on a policy issue deemed relevant to that PAC.* In other words, these are cases where a donor donates substantially more than they usually donate in that congressional cycle and these donations are significantly more likely to occur in close proximity to a related floor speech (which are speeches that prioritize the issue of the donor's own interest). The method can also find the individual legislators driving this significant connection and thus surface cases of legislator-PAC relationships for further investigation.

We show one such example in Table 1—easily validated by using a search query of the form '<PAC Name> <Legislator Name>' in a search engine. The complete procedure and some findings are provided in S10 Appendix, which include examples of significant proximate speech and donation events (S7 Table) and additional examples of legislator-PAC relationships (S8 and S9 Tables). Taken together, our procedures and illustrative examples demonstrate the potential for increased public transparency over and above what raw donation transaction data or anecdotal evidence alone can provide.

## Discussion

We study the connection between PAC donations to elected legislators and the issues those legislators prioritize in their speeches on the floor of Congress. This helps shed new light on the association between organized donations and the behavior of elected representatives, while also increasing the understanding of PAC donations as a factor in studies of issue salience or *attention* in Congress. The research here expands the ways in which we think about the potential impact of donations on policymaking and political representation.

Our key finding is that information about which PACs did and did not donate to a US House representative is meaningfully associated with representatives' issue-attention derived from their floor speeches. Recognizing and quantifying this association is important for understanding the dynamics of political power as exercised through agenda setting and control in Congress. We find that donors (PACs) are significantly more associated with legislators' issue-attention than other conventional factors (such as committee assignments or partisanship). This association is made clearer in two different ways: first, the relationship holds for each congressional cycle over a period of twenty-four years, demonstrating that our findings are based in a consistent pattern for the US House of Representatives; and second, our predictive machine learning model can automatically learn meaningful issue-PAC associations,

highlighting the strong patterns of PACs donating to those legislators who prioritize issues relevant to the PACs' own policy area of operation.

Our work focuses on analyzing the association between issue-attention and explanatory variables such as PACs and does not make claims about causality. We cannot say whether the relationship reflects donors rewarding legislators for their support on the House floor, or whether donations spur a legislator to talk about an issue more than they would have otherwise, or whether both of these dynamics are happening. However, what our study uniquely shows is that there is a clear positive relationship between who gives money to politicians and what those elected officials talk about. This is consistent with research on the relationship between money and votes that also finds an association, but not necessarily causality. Establishing this positive relationship between money and speech is a critical first step toward a better understanding of the important relationship between donors and legislators. With our novel focus on issue priorities expressed in language (as opposed to roll-call votes), combined with our methodology and the new data presented, future research is better positioned to examine the age-old question of causality with new tools. Future work should explore legislators who 'break character' by talking about an issue they generally do not pay attention to in a particular period of time, and systematically analyze if donors interested in that issue tend to donate to the legislator before or after those unexpected speeches. This could imply an attempt to influence the legislator, or a reward for the difference in activity, respectively.

We acknowledge the possibility of other explanatory variables not considered in this study that could potentially predict legislators' issue-attention better than information about their donors. However, since donor information offers significantly more predictive power than standard explanations, does so consistently over time, *and* constitutes a strong enough pattern that the automatic discovery of meaningful issue-PAC associations is possible, our results present a useful finding and a foundation on which future work can build. Future studies can consider other variables or attributes such as legislators' race and gender, or attributes of their districts.

The release of our new database, our codebase, annotations of topic modeling output (to get issue labels), ratings for issue-PAC associations, and other procedures and modeling outputs are intended to help spur future research into both the impact of donations on congressional activity as well as possible factors explaining agenda setting on the floor of Congress. One particular avenue for future work is to consider the various aspects and stances within a political issue present in floor speeches given by a legislator and move beyond *issue-attention* to *framing*. Now that we know that donors are meaningfully associated with how much legislators talk about an issue, the next question can be: are they also associated with legislators' stances and framing choices in the context of a particular issue?

Another avenue for future research is to consider a different aspect of agenda-setting: what is *not* being said, as opposed to what is being said. From a broader theoretical perspective, the issue of non-events (such as what is not said) is an important consideration rooted in classic work on power and politics [58, 92–94]. Our current methodology does not support modeling legislators who do not engage in any floor speech activity. Our dataset, however, can help drive research with models that do not have this limitation and can effectively study those legislators who do not give floor speeches. In this work, we only consider legislators active in terms of giving floor speeches and their presence in Congress across multiple congressional sessions in our processed data (exact thresholds and details provided in S2 Appendix). However, our approach *does* model the complete topic distribution or issue-attention, and thus considers legislators not speaking about some issues at all while they prioritize other issues. Incorporating and reflecting legislators' choices to talk about some issues and not others, our approach examines the extent to which these choices are related to who they receive money from. Future

work is therefore poised to further investigate and model what is *not* said on specific political issues. Such future work can better understand the impact of various sources of information about legislators, such as their donors, on this other aspect of agenda-setting.

Apart from scholarly work, our data and identification of issues and PACs annotated as related to those issues can serve as the core of an *enabling technology*. It can help journalists identify donation and speech events in Congress that deserve scrutiny. Specifically, we contribute a procedure that uses our trained machine learning model outputs and expert judgments of issue-donor associations to help find significantly-connected speech and donation events occurring in close proximity to one another. Identifying such pairs of events can help uncover significant PAC-legislator relationships, such as the example provided in Table 1. Examples of connected speech and donation events along with more examples of PAC-legislator relationships discovered using our approach are provided in S7–S9 Tables.

## Supporting information

**S1 Appendix. Detailed data overview.**
(PDF)

**S2 Appendix. Data processing.**
(PDF)

**S3 Appendix. Distribution of donations across legislators.**
(PDF)

**S4 Appendix. Human issue-PAC association ratings task.**
(PDF)

**S5 Appendix. Results with issue-attention derived from substantive policy issues alone.**
(PDF)

**S6 Appendix. Training details.**
(PDF)

**S7 Appendix. Statistical testing.**
(PDF)

**S8 Appendix. *PAC* versus *Committee* legislator attributes.**
(PDF)

**S9 Appendix. Results with combinations of various legislator attributes.**
(PDF)

**S10 Appendix. Procedure for surfacing potential temporal connections.**
(PDF)

**S11 Appendix. Robustness check for a potential dependency of our finding on the choice of topic modeling method used to obtain legislators' issue-attention by using three different topic models other than LDA.**
(PDF)

**S12 Appendix. Robustness check for a potential dependency of our finding on the order of documents used to train the LDA topic model.**
(PDF)

**S13 Appendix. Robustness check for a potential dependency of our finding on the number of topics used to train the LDA topic model.**
(PDF)

**S14 Appendix. Detailing and comparing different types of PACs in our processed dataset: Business PACs offer a significantly higher association with issue-attention than labor PACs.**
(PDF)

**S15 Appendix. Details about constructing two additional legislator attributes—*Seniority* (number of congressional terms served) and *District Marginality* (how close their district is in voting in US presidential elections)—And discussing comparison with other attributes at the level of specific congressional sessions.**
(PDF)

**S1 Fig. Our new dataset's schema.** Note that not all data columns are shown for each table (relation), and the schematic aims to simply provide an overview of the structure and contents of our database. In this database we have created, the PAC contributions data runs from 1989–2018, while the floor speeches are from 1994–2020. There are 5 possible values for the **type** of PAC: Business, Labor, Ideological, Other, and Unknown—we only consider Business and Labor PACs in our study.
(TIF)

**S2 Fig. PAC donation sizes in terms of the number of different legislators (as recipients).** In our processed dataset used to train our machine learning framework with 758 legislators who are present and give floor speeches across 1995–2018, and 1002 PACs actively donating in the same period; most PACs tend to donate to less than a third of all the legislators they could donate to.
(TIF)

**S3 Fig. Donation patterns across political parties.** In our processed dataset used to train our machine learning framework, most PACs tend to donate across party lines instead of donating in an exclusive, partisan manner.
(TIF)

**S4 Fig. Comparison for all legislator attributes after removing non-substantial non-policy issues.** We find the same pattern as when using all topics: PAC attribute explains legislators' issue-attention the most.
(TIF)

**S5 Fig. Held-out set results with error bars when modeling each congressional session separately and predicting legislators' issue-attention.** These correspond to the held-out set results (without error bars) shown in Fig 2B.
(TIF)

**S6 Fig. Training set results when modeling the entire dataset and predicting legislators' issue-attention.** Corresponding held-out set results are shown in Fig 2A.
(TIF)

**S7 Fig. Training set results when modeling each congressional session separately and predicting legislators' issue-attention.** Corresponding held-out set results are shown in Fig 2B.
(TIF)

**S8 Fig. Specific comparison for the *PAC* and *Committee* legislator attributes.** While *PAC* offers more association with issue-attention, results for the combined attribute set of *PAC* and *Committee* suggest some complementary information present in these two explanatory variables. (TIF)

**S9 Fig. Specific comparison for the *PAC* and *Committee* legislator attributes—When modeling each congressional session separately.** (TIF)

**S10 Fig. Comparison for all legislator attributes including combinations of *PAC* with non-donor attributes.** Only *Committee* information increases the explanatory power of the *PAC* attribute. (TIF)

**S11 Fig. Example of how the rating task is presented to an expert.** Given their own topic label, and 10 PACs, an expert rated the association of the PAC with the topic on a 1–3 Likert scale. (TIF)

**S12 Fig. Issue-PAC associations discovered by the model are deemed meaningful by experts.** Additional results of experts rating issue-PAC associations, showing that the top PACs per issue as per our regression model's weights are meaningful, since the model's top 10 PACs are rated consistently higher in terms of association with the issue as compared with a random selection of 10 PACs for the issues. A value of 5 (as an example) on the x-axis here means *at least* 5 out of the 10 PACs shown were rated as clearly associated with the issue (a rating of 3). More PACs were rated as related to the issue, out of the 10 shown, when the PACs shown were selected based on the associations learned by our model compared with a random selection. (TIF)

**S13 Fig. Algorithmic workflow of identifying relevant issue-PAC associations that show cases of significantly connected speech and donation events.** These are cases where, within a short time window (proximal), a donation made by a PAC interested in a particular issue and a speech made by the recipient on that issue co-occur. (TIF)

**S14 Fig. Results comparing our four main legislator attributes when issue-attention is obtained using the Structured Topic Model or STM [86, 87].** (TIF)

**S15 Fig. Results comparing our four main legislator attributes when issue-attention is obtained using Non-negative Matrix Factorization or NMF [88–90].** (TIF)

**S16 Fig. Results comparing our four main legislator attributes when issue-attention is obtained using the Contextualized Topic Model or CTM [91], where the topic model now uses both the terms in the input collection of floor speeches represented as a bag-of-words and the same input collection represented using contextual embeddings for sentences generated using BERT, a large pre-trained language model [95, 96].** (TIF)

**S17 Fig. Results comparing our four main legislator attributes across ten LDA-based topic modeling runs, each trained using a different (random) order of the input speeches or**

**documents.**
(TIF)

**S18 Fig. Results comparing our four main legislator attributes when LDA is trained using a different number of topics ($K$ = 30) than our main experimental results shown in Fig 2A.**
(TIF)

**S19 Fig. Results comparing our four main legislator attributes when LDA is trained using a different number of topics ($K$ = 45) than our main experimental results shown in Fig 2A.**
(TIF)

**S20 Fig. Results comparing our four main legislator attributes when LDA is trained using a different number of topics ($K$ = 90) than our main experimental results shown in Fig 2A.**
(TIF)

**S21 Fig. Results comparing our four main legislator attributes when LDA is trained using a different number of topics ($K$ = 120) than our main experimental results shown in Fig 2A.**
(TIF)

**S22 Fig. Results comparing the association with issue-attention offered by different types of PACs, by considering one particular type of PAC (business or labor) when representing legislators using their donor profile.**
(TIF)

**S23 Fig. Results comparing the association with issue-attention offered by various legislator variables including *Seniority* on the held-out set when models are trained separately for each congressional session in our data (direct addition of one legislator attribute to results presented in Fig 2B).**
(TIF)

**S24 Fig. Results comparing the association with issue-attention offered by our four main legislator attributes as well as a legislator attribute capturing how close (or marginal) a legislators' district is in the recent presidential election (called *District Marginality*).** The comparison is done across three different Congresses—2009–10, 2013–14, and 2017–18—and uses data on district-level voting for US presidential candidates immediately preceding the particular Congress under consideration.
(TIF)

**S1 Table. Dimensions or sizes of the various legislator attributes.** The attributes are used to predict the 758 X 60 legislator-level average topic proportions in our multinomial regularized logistic regression approach.
(PDF)

**S2 Table. The dimensionality of each of our legislator attributes for the modeling done separately for each congressional cycle.** The first value in each dimensionality is the number of speakers selected in that cycle, and the second value is the number of features (for the legislator attribute or model indicated in the column title). Note that the y for each of the regression models (averaged topic distribution for speakers) is going to be (number of speakers, 30).
(PDF)

**S3 Table. Examples of expert-provided issue labels for topics.** Note that experts created labels for topics using not just the top terms, but also the top documents (floor speeches) for the topic given by the topic model. And contextualizing info such as sector, industry, and the

industrial category was provided when asking for association ratings on the 1–3 Likert scale.
(PDF)

**S4 Table. Values for optimal hyperparameter settings for the multinomial logistic regression model.** The best settings found for *PAC* were also used for *Random-PAC*.
(PDF)

**S5 Table. Statistical significance test results using Mann-Whitney *U* test, comparing 30-fold cross-validation results for *PAC* against other legislator attributes.** $p < 0.05$ (bold-faced) indicates significantly better results for the *PAC* attribute against the other attribute for that congressional session.
(PDF)

**S6 Table. Statistical significance test results using Mann-Whitney U test, comparing 30-fold cross-validation results for *Committee* against other legislator attributes.** $p < 0.05$ (boldfaced) indicates significantly better results for the *Committee* attribute against the other attribute for that congressional session.
(PDF)

**S7 Table. Examples of meaningful (non-random) relevant speech and donation events occurring in close proximity.** After selecting relevant issue-PAC pairs with potential temporal connection for the cycle: some examples of speeches made on the topic and cases of the relevant PAC donating an amount much higher than their mean donation amount during that congressional cycle to the legislator giving that speech within a particular time window around the speech.
(PDF)

**S8 Table. An example of the kind of real-world relationships that the outputs of our modeling and analysis can capture automatically.** For a PAC (BAE Systems) and an issue (International Security/Foreign Policy) that our model (as well as experts) deem related to one another, we find that in a particular congressional cycle (1995–96), there is a significant temporal connection between this PAC and the issue, *i.e.*, this PAC donates a significant amount closer to speeches on the particular issue of international security and foreign policy in 1995–96 than for speeches that are not on this issue. In particular, the legislator—Norm Dicks (D-WA)—emerged as a frequent recipient of such temporally significant donations by this PAC, and a simple search on a search engine reveals that this connection is backed by what is known to journalists and validated by real-world knowledge in a straightforward manner (the last three columns of the table).
(PDF)

**S9 Table. An example of the kind of real-world relationships that the outputs of our modeling and analysis can capture automatically.** Or a PAC (National Assn of Federally Insured Credit Unions) and an issue (Finance/Financial regulation) that our model (as well as experts) deem related to one another, we find that in two different congressional cycles (2009–10 and 2015–16), there is a significant temporal connection between this PAC and the issue, *i.e.*, this PAC donates a significant amount closer to speeches on the particular issue of finance or financial regulation in both 2009–10 and 2015–16 than for speeches that are not on this issue. In particular, the legislator—Ed Royce (R-CA)—emerged as the recipient of such temporally significant donations by this PAC in both cycles, and a simple search on a search engine reveals that this connection is backed by what is known to journalists and validated by real-world knowledge in a straightforward manner (the last three columns of the table).
(PDF)

## Acknowledgments

We thank Alexander Hoyle, Stefan McCabe, Jon Green, Dallas Card, Sarah Shugars, David Lazer, Jillian Rothschild, Patrick Wohlfarth, the participants of the University of Maryland's American Politics workshop, and the members of the University of Maryland's Computational Linguistics and Information Processing Laboratory for their helpful comments.

## Author Contributions

**Conceptualization:** Pranav Goel, Nikolay Malkin, SoRelle W. Gaynor, Nebojsa Jojic, Kristina Miler, Philip Resnik.

**Data curation:** Pranav Goel.

**Formal analysis:** Pranav Goel.

**Funding acquisition:** Nebojsa Jojic, Kristina Miler, Philip Resnik.

**Investigation:** Pranav Goel.

**Methodology:** Pranav Goel.

**Resources:** Nebojsa Jojic, Philip Resnik.

**Software:** Pranav Goel.

**Supervision:** Nikolay Malkin, SoRelle W. Gaynor, Nebojsa Jojic, Kristina Miler, Philip Resnik.

**Validation:** Pranav Goel, SoRelle W. Gaynor, Kristina Miler.

**Visualization:** Pranav Goel.

**Writing – original draft:** Pranav Goel.

**Writing – review & editing:** Pranav Goel, Nikolay Malkin, SoRelle W. Gaynor, Nebojsa Jojic, Kristina Miler, Philip Resnik.

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
