## [Decision Letter · Decision Letter 0]

26 Apr 2023

PONE-D-23-06208Donor activity is associated with US legislators’ attention to political issuesPLOS ONE

Dear Dr. Goel,

Thank you for submitting your manuscript to PLOS ONE. After careful consideration, we feel that it has merit but does not fully meet PLOS ONE’s publication criteria as it currently stands. Therefore, we invite you to submit a revised version of the manuscript that addresses the points raised during the review process. Please address reviewers' and academic editor's comments and suggestions accordingly.

We look forward to receiving your revised manuscript.

Kind regards,

Yongjun Zhang

Academic Editor

PLOS ONE

“We thank Alexander Hoyle, Stefan McCabe, Jon Green, Dallas Card, Sarah Shugars,

David Lazer, Jillian Rothschild, Patrick Wohlfarth, the participants of the University of

Maryland’s American Politics workshop, and the members of the University of

Maryland’s Computational Linguistics and Information Processing Laboratory for their

helpful comments. P.G., S.W.G., K.M., and P.R. were supported by the National

Science Foundation (NSF) under Grant 2008761. P.G. and P.R. were also supported by

the NSF Grant 2031736 and by Amazon. P.G. was additionally supported by an

internship at Microsoft Research.”

“National Science Foundation grant 2008761 (PG, SWG, KM, PR) (https://www.nsf.gov/awardsearch/showAward?AWD_ID=2008761)

National Science Foundation grant 2031736 (PG, PR) (https://nsf.gov/awardsearch/showAward?AWD_ID=2031736)

Amazon (PG, PR) (https://www.amazon.science/research-awards/program-updates/2020-amazon-research-awards-recipients-announced)

3. We note that Figures 4, S14 and S15 in your submission contain copyrighted images. All PLOS content is published under the Creative Commons Attribution License (CC BY 4.0), which means that the manuscript, images, and Supporting Information files will be freely available online, and any third party is permitted to access, download, copy, distribute, and use these materials in any way, even commercially, with proper attribution. For more information, see our copyright guidelines: http://journals.plos.org/plosone/s/licenses-and-copyright.

a. You may seek permission from the original copyright holder of Figures 4, S14 and S15 to publish the content specifically under the CC BY 4.0 license.

b.If you are unable to obtain permission from the original copyright holder to publish these figures under the CC BY 4.0 license or if the copyright holder’s requirements are incompatible with the CC BY 4.0 license, please either i) remove the figure or ii) supply a replacement figure that complies with the CC BY 4.0 license. Please check copyright information on all replacement figures and update the figure caption with source information. If applicable, please specify in the figure caption text when a figure is similar but not identical to the original image and is therefore for illustrative purposes only.

Additional Editor Comments:

After carefully reading your paper in consultation with two reviewer's comments, I am happy to extend a Major Revision. Please address those two reviewers' comments accordingly. Also, related to topic modeling, reviewer 1 mentioned several alternative ways on top of LDA. I would love to see the robustness checks using STM or BERTopic. Regarding machine learning experimental framework, I was wondering if you can do some robustness checking using word embedding approach. For instance, you can have a set of "issues"- a dictionary of words or n-grams representing a concept and then you measure the distance or similarity between the concept and speech.

Reviewers' comments:

Reviewer's Responses to Questions

**Comments to the Author**

1. Is the manuscript technically sound, and do the data support the conclusions?

Reviewer #1: Yes

Reviewer #2: Yes

2. Has the statistical analysis been performed appropriately and rigorously? 

Reviewer #1: Yes

Reviewer #2: Yes

3. Have the authors made all data underlying the findings in their manuscript fully available?

Reviewer #1: Yes

Reviewer #2: Yes

4. Is the manuscript presented in an intelligible fashion and written in standard English?

Reviewer #1: Yes

Reviewer #2: Yes

5. Review Comments to the Author

Reviewer #1: This article is quite interesting and well formulated. The use of NLP methods to connect congressional ideology to donor information is novel. Analyzing the content of congressional speeches in this effort is appropriate. The article, as written, is quite easy to follow by a generalist audience. The contribution is strong. There are a few issues that I would recommend that the authors take care of. Thus I recommend that this article receive a minor revision as I really only have quibbles with the topic modeling procedure.

- On pgs. 3-4 the authors repeat a variation of the phrase "includes information about legislators, such as their

state, party affiliation, and committee assignments..." etc. three times. Simply introduce the type of metadata that the authors include one time.

- On pg. 5 there is a link to Navient's wikipedia page that seems erroneous. I suspect that this is a footnote gone wrong using LaTeX.

- The authors use LDA topic modeling procedures to identify the underlying themes, but as it stands now, the justification for this is simply insufficient. LDA can be useful, particularly in a comparative context for standardizing estimation across different corpuses. However, there are other topic modeling procedures that have been developed, specifically to include metadata in the extraction process (Structural Topic Models (STM)), or to reduce dimensionality (Negative Matrix Factorization (NMF)). One of the problems with LDA is that the order of the training dataset biases the output. The authors provide no information about how the training data was ordered which is important for interpreting whether LDA finds anything meaningful. One could think of many feasible ways to do this which complicate this methodological choice. This could be done by year, by order of delivery on the floor, within and between parties etc. The authors need to justify this decision, and to be honest, I think a robustness check with NMF is in order if LDA is settled upon as the appropriate technique after working through this puzzle. LDA might be the preferred model by practitioners, but that may have a lot to do with the fact that prior to recent computational advances, LDA was really the only model used by practitioners. This is not the case now and readers of PLOS One will expect a more cogent argument for this selection. None of this information is provided in the existing appendices.

- How do the authors select the number of topics? This seems especially important since the posterior distribution is averaged across speeches to generate a measure of issue-attention. Some topic modeling procedures have built in methods (e.g., SearchK function in STM) but with LDA you have to demonstrate that the number of topics that you select balance coherence against exclusivity through some set of researcher generated procedures. The authors provide little in the way of this information. None of this information is provided in the existing appendices.

- I'm not an expert in ML optimization techniques so I will defer to the other reviewers on this subject.

Reviewer #2: This is an interesting and creative paper that uses a variety of new computational methods to detect an association between PAC contributions and congressional floor speech. The authors find that legislators’ “issue attention” is strongly predicted by which PACs contributed to that legislator, even net of other legislator attributes. This is surely an important finding for social scientists and will be of interest to a broad range of scholars.

My major concerns are with the framing of the paper and the authors’ claims to novelty:

• First, the introduction contains a few mischaracterizations that will raise the hackles of scholars of campaign finance. The authors state: “Most US-based corporations, unions, and interest groups organize a Political Action Committee (PAC) to raise funds…” In a well-known article, Ansolabehere et al. (2003) find that about 60% of Fortune 500 companies (i.e., large, publicly traded corporations) have PACs. If we expand the universe to all U.S.-based corporations (including public and private?), the percentage is likely much, much lower. I have never seen a precise estimate of the proportion of unions/interest groups (what kind of interest groups?) that have PACs. The authors should revise/clarify/qualify this statement to correspond to what we actually know about PACs, i.e. a majority of publicly-traded corporations have PACs, but the percentage is far from “all” corporations and not exactly “most.”

Ansolabehere, S., De Figueiredo, J. M., & Snyder Jr, J. M. (2003). Why is there so little money in US politics?. Journal of Economic perspectives, 17(1), 105-130.

• Second, the authors assert: “Research has found a generally positive correlation between donor activity [what kind?] and PAC influence [defined how?] on roll call voting for specific policy initiatives.” The authors then say: “broadly there is conflicting evidence of whether contributions influence roll call votes” (2). Could the authors clarify what they mean by these seemingly contradictory statements? In my reading of the literature, the former statement is not correct. Much of the literature is a debate between scholars who find such an association and those who find no such association. Stratmann (2005) provides a helpful review:

Stratmann, T. (2005). Some talk: Money in politics. A (partial) review of the literature. Policy challenges and political responses: Public choice perspectives on the post-9/11 world, 135-156.

• I do think the authors are making an important and novel contribution to the literature, but the findings should be better contextualized vis-à-vis recent work. In this way, the manuscript is adding to other recent efforts to “move beyond the focus on votes.” For instance, Hertel-Fernandez has shown that policymakers include verbatim sections of model bills authored by interest groups in draft legislation:

Hertel-Fernandez, A. (2019). State capture: how conservative activists, big businesses, and wealthy donors reshaped the American states--and the nation. Oxford University Press, USA.

Finally, I appreciated the authors’ humility vis-à-vis causality by explicitly referring to their estimates as ones of association. However, I have a few comments/suggestions for the methods section to make this paper more relevant to past literature:

• The authors only observe one portion of the agenda-setting process by modeling floor speech. However, scholars have long been concerned with what is *not* said as much as what is said. In other words, the authors are modeling legislators’ issue attention conditional on speaking, but could the authors say something about the “selection” / first stage in this two-step process? That is, what would we find if we tried to model / include the 0s for legislators who simply do not engage in floor speech?

• Political action committees are a very wide category of political committee that contains two separable types of organizations: 1) separate segregated funds/connected and 2) nonconnected. The former category generally includes corporate, trade, and labor PACs while the latter mainly includes ideological groups. I understand the authors are “controlling” for the actual PAC in their models, but I wonder if it would be more interesting to parse some of this variation by including greater granularity vis-à-vis PAC characteristics. For instance, the authors could include “PAC type” in addition to industry / category (or at least explain how this broadly recognized classification relates to their coding).

• Similarly, it might be helpful to include legislator control variables such as: years in Congress, party leadership, marginal district, etc.

• The authors discuss the possibility of reverse causality in the conclusion, but the estimates are vulnerable to other forms of endogeneity like omitted variable bias. How would including variables such as those mentioned above likely impact their estimates?

6. PLOS authors have the option to publish the peer review history of their article (what does this mean?). If published, this will include your full peer review and any attached files.

Reviewer #1: No

Reviewer #2: No

---

## [Author Response · Author response to Decision Letter 0]

11 Jul 2023

Dear Editor Zhang,

Thank you for the thorough and constructive comments on our manuscript, “Donor activity is associated with US legislators’ attention to political issues”, and for extending the invitation to revise and resubmit our manuscript for consideration as a research article in PLOS ONE. We believe that our work and the manuscript are significantly enhanced by the feedback provided by yourself and the reviewers. 

We appreciate that both reviewers consider our study interesting and our methodology novel and creative. Reviewer 1 notes the clear strength of our contribution while Reviewer 2 also notes the importance of our finding for social scientists. Reviewer 1 suggests minor revisions that take into account some important robustness checks around our topic modeling procedure – we comprehensively address each comment with new experiments and robustness checks as suggested. Reviewer 2 notes issues with framing, limitations in proper contextualization of our work and how it is situated in the literature, and also notes other considerations in methodology. We have adjusted our framing and again perform new experiments and checks to incorporate this valuable feedback. 

Below we provide our detailed response and discuss the changes we make in our manuscript in order to address each comment provided by the editor and both reviewers: 

Robustness checks on topic modeling-related decisions

Both the Editor and Reviewer 1 suggested we consider additional robustness checks with respect to our topic modeling-related decisions, pointed out other existing topic modeling procedures to consider, and recommended providing more discussion about our use of LDA.

We agree that it is beneficial to establish the reliability of our findings by conducting a robustness check with different topic modeling methods. In order to establish the robustness of our main findings with respect to the choice of topic modeling method, we follow recommendations from Reviewer 1 (and the Editor) and present our main result with three different topic modeling methods: STM, NMF, and the Contextualized Topic Model (CTM), which is a neural topic modeling method that uses contextualized word embeddings (Bianchi et al., 2021; this last model is also discussed below in response to feedback from the editor). Crucially, we have confirmed that our main finding, based upon LDA-derived topics, continues to hold when the three alternative topic modeling methods are used: PACs are significantly more associated with legislators’ issue-attention in floor speeches than other legislator attributes. Details for the implementation used and discussion of the results are provided in S11 Appendix, with the main results comparing PAC, Committee, State, and Party presented in Figures S14-S16 in the revised manuscript. 

We also agree with the reviewers' recommendation to include a more expansive discussion of the choice to use LDA, especially in light of recent computational advances and new approaches in topic modeling. We have revised the manuscript to more explicitly make the case for using LDA (page 7 in the marked-up version of our revised manuscript) in addition to the aforementioned new robustness checks with different topic modeling methods. For example, prior work in Hoyle et al. (2021) has found that classical LDA is the dominant topic model of choice among practitioners and that LDA yields strong qualitative ratings for its topics as judged by humans. Recent advances in the machine learning and natural language processing literature have introduced new types of topic models, with one such noteworthy introduction being neural topic models that utilize deep neural networks (Srivastava et al., 2017). However, these new topic models claimed improvement over LDA via an automatically computed measurement that has since been established as an invalid proxy of human judgment in Hoyle et al. (2021). Further, Hoyle et al. (2022) establish that LDA is a more reliable choice for qualitative content analysis than recent neural topic modeling methods. Taken together, Hoyle et al. (2021) and Hoyle et al. (2022) make a strong case for LDA as the choice of topic modeling method for qualitative content analysis in order to interpret latent categorical themes present in a document collection (containing English text). 

Reviewer 1 also notes that one of the problems with LDA is that the order of the training dataset can bias the output. 

We appreciate Reviewer 1 pointing this consideration out and acknowledge that, technically, our Gibbs sampling-based LDA approach cannot be expected to give the same topic modeling output for different orders of the training dataset, though we expect similar outputs. 

In order to ensure that our main finding about PACs offering the strongest association with issue-attention holds regardless of training data order, we conduct an additional robustness check in which we randomly shuffle the order of the training dataset 10 times. We find that our results are replicated in all of these runs. Figure S17, added in the revised manuscript, shows that the obtained values of bit reduction (%) are stable across the ten trained LDA models. This new robustness check is discussed in the S12 Appendix of our revised manuscript. 

These results confirm that the order of the training dataset does not affect the findings of our work. Robustness checks with other topic modeling methods, including NMF (as suggested), also replicate our finding (S11 Appendix; Figure S15 in our revised manuscript). 

We also point to the new S12 Appendix in our marked-up revised manuscript in footnote 15 on page 8. 

Reviewer 1 asks for more detail on how we select the number of topics for our analysis. As they point out, this could be particularly important given the posterior distribution is averaged across speeches to generate a measure of issue-attention. 

This is an important issue and we appreciate the note that greater clarity was needed. In our work, we do not just rely on the automatically obtained topic distribution (using LDA) for our main finding (Figure 2A). We conduct an extensive expert curation process to obtain a coherent set of labels for political issues used in our analyses (step 1 in the human curation procedure discussed in S4 Appendix). We show that after removing non-coherent or non-substantive categories or issues, our main finding continues to hold (S5 Appendix and S4 Fig): PACs offer the strongest association with issue-attention. The initial number of topics (60 in our work) is reduced to a set of identified categories based on expert judgment (48 in our work). We believe this is an important way to establish the reliability of our results, since our finding holds for both the initial set of automatically discovered topics and the expert-refined or curated set of issues. 

Our submitted manuscript did not make this important curation effort sufficiently clear, and the resulting check sufficiently explicit. In our revised manuscript, this step is more clearly highlighted in the main text towards the end of page 7 (marked-up version). 

Furthermore, recognizing that different practitioners might rely on a different number of topics for their initial estimates, we have conducted an additional robustness check where we derive issue-attention using different values for the number of topics (K). In our submitted manuscript, we set K = 60. We now show our main finding also holds if we instead use 30, 45, 90, or 120 as the value for K. This shows that our finding is robust across a range of different values for the number of topics. We discuss this robustness check in the newly added S13 Appendix and visually present findings for each of these new values of K in Figures S18-S21 in our revised manuscript. We point to S13 Appendix in our main paper in footnote 18 on page 9 of our marked-up revised manuscript. 

Use of word embeddings

Regarding the machine learning experimental framework, the editor requests a robustness check using a word embedding approach. 

We agree with the editor that the bag-of-words representation used in the topic modeling framework used in our submitted manuscript has limitations. Making use of embeddings that exploit the sequential realization of the fundamentally hierarchical structure of the English language via the context of each word or term in the data would be valuable to incorporate as a robustness check. To this end, we show that our main finding holds if we use the Contextualized Topic Model or CTM (Bianchi et al., 2021) instead of LDA (Figure S16; details provided in the S11 Appendix in our revised manuscript). 

CTM uses a combined representation of documents: bag-of-words as well as documents embedded using a large, pre-trained language model — BERT (Devlin et al., 2019) — which has been shown to offer powerful contextualized word representations. Sentence BERT or SBERT (Reimers et al., 2019) is used to derive document embeddings using BERT in the CTM method. This robustness check uses word embeddings and is in keeping with some of the recent advances in both word embeddings and topic modeling. 

With respect to other clustering and embedding based approaches, it is also important to note that in our work, issue-attention is conceptualized as a probability distribution over political issues. Our machine learning framework, including the loss function used for training and the information theory-based measurement used for evaluation, rely on this conceptualization. Similarities in embedding space do not have the same distributional properties and do not sustain the mixed-membership nature of topic distributions. We also note that the aforementioned prior work in Hoyle et al. (2022) compares LDA with multiple topic modeling techniques that make use of word embeddings and establishes LDA as the superior choice in terms of reliable use for qualitative content analysis. 

Adjust framing to contextualize and better account for prior work in the introduction and substantive setup for our work

Reviewer 2 notes that the introduction contains a few mischaracterizations of campaign finance, namely our estimation of the percentage of U.S.-based corporations that have a Political Action Committee. Overall, they recommend that we clarify and/or qualify our statement to correspond to what we actually know about PACs (i.e. a majority of publicly-traded corporations have PACs, but the percentage is far from “all” corporations and not exactly “most”).

We appreciate the Reviewer’s close reading and correction. In accordance with their recommended modification, we have changed the first line of our manuscript (page 1 in the revised marked-up version) from “Most US-based corporations, unions, and interest groups organize a Political Action Committee (PAC) to raise funds…” to now read: “A majority of publicly-traded US-based corporations as well as many labor unions and interest groups organize a Political Action Committee (PAC) to raise funds…”. This clarification should provide a more accurate picture and characterization to our readers. 

Reviewer 2 requests that we clarify our discussion of the existing literature, particularly our characterization of the evidence regarding PAC influence on congressional behavior as unresolved/inconclusive/up for debate. They also point to Stratmann (2005) for a helpful review on the debate on association (or lack thereof) between donor activity and roll call voting. 

Reviewer 2 also notes that while our work makes an important and novel contribution to the literature, our findings should be better contextualized vis-à-vis recent work. They highlight that our manuscript is adding to other recent efforts to “move beyond the focus on votes”, and point to relevant recent work by Hertel-Fernandez.

We thank the reviewer for the suggestion to better situate our work in the existing literature, and agree that in our effort to be concise, we omitted some important references and were not sufficiently clear about the state of the literature. Together with the reviewer's helpful suggestions (particularly Stratmann et al. 2005), we have worked to expand this discussion. 

Our intent in the original manuscript was to convey that the literature provides some evidence of correlation, but conflicting evidence of causal influence. However, we agree with the reviewer that the discussion was unclear. We have updated the text to reflect the fact that research comes to mixed conclusions regarding both causation and correlation between donations and roll-call votes, also incorporating the reference to Stratmann (see page 2, paragraph 2 of the marked-up revised manuscript). 

We added new language that highlights the limitations of voting data and the advantages offered by using language data, specifically floor speeches, in the third and fourth paragraphs of our introduction (across pages 2 and 3 in the marked-up revised manuscript). We also added a number of relevant citations as well (in addition to the relevant work by Hertel-Fernandez recommended by the Reviewer), including (but not limited to) recent work by Iliev (2021) who examines committee discussions on energy policy and finds evidence of a bidirectional relationship between donors and U.S. senators, and Vallejo Vera (2021) who finds evidence of strategic use of interest groups' committee speech to influence legislators' policy positions (see page 3 in the revised marked-up manuscript). Finally, we also now acknowledge the more consistent evidence that money affects non-voting legislative behavior (page 3 of revised marked-up manuscript) based on a body of prior work (Hall and Wayman 1990; Esterling 2007; Powell and Grimmer 2016). 

We think that this revised discussion nicely motivates our choice to focus on language use and highlights the contribution of our finding of robust association, and we again thank the reviewer for suggesting these changes. 

Method-related edits and expanded discussion/ substantive considerations in methodology

Reviewer 2 notes that in our work, we only observe one portion of the agenda-setting process by modeling floor speech, but scholars have long been concerned with what is *not* said as much as what is said. In other words, we are modeling legislators’ issue attention conditional on speaking. The reviewer asks us if we could consider the legislators who simply do not engage in floor speech.

This is a very interesting point raised by Reviewer 2, and we acknowledge that this work models legislators' issue attention conditional on speaking. From a broader theoretical perspective, the issue of non-events (such as what is not said) is an important consideration rooted in classic work on power and politics (e.g., Dahl 1956, Bachrach and Baratz 1962, 1963; Gaventa 1980). And we share the reviewer's perspective that what is not said is also interesting. 

As the reviewer notes, the current work excludes legislators who do not speak on any issue at all. In our framework, the topic modeling approach quantifying our issue-attention as a distribution requires floor speeches from legislators, and cannot directly incorporate this other portion of the agenda-setting process. In other words, without floor speeches, we cannot quantify issue-attention for a legislator as things are currently operationalized, given our machine learning framework cannot handle all 0s in a row (corresponding to a legislator), as it breaks required properties for our dependent variable, such as it being a probability distribution. 

We do note, however, that our approach does model the complete issue distribution, and therefore it does consider legislators not speaking about some issues at all while they prioritize other issues (0 or close to 0 values for some issues for legislators). This means that our model does incorporate and reflect legislators' choices to talk about some issues and not others, and examines the extent to which this is related to who they receive money from. Future work is therefore poised to further investigate and model what is not said on (at least) specific political issues using our data and methodology. We have added text to clarify this in footnote 16 in our revised manuscript (marked-up version). 

In future work, we hope to directly model the number of floor speeches given by a legislator as the dependent variable, and assess if donors and other attributes can predict floor speech activity including no floor speech activity specifically. We agree that modeling such legislators could lead to interesting insights into the other portion of agenda-setting, and we hope our dataset can enable this research with a different framework and methodology. We now note this is both a limitation and a potentially rich avenue for future research in our updated discussion section (the last paragraph beginning on page 14 in the marked-up revised manuscript). 

Reviewer 2 points out that Political Action Committees is a broad term that includes different types of organizations, and notes that while we are “controlling” for the actual PAC in their models, they wonder if it would be more interesting to parse some of this variation by including greater granularity vis-à-vis PAC characteristics.

We thank the reviewer for pointing out that we need to be clearer in discussing the types of PACs included in this research, and for the suggestion to explore whether there are differences between the types of PACs. In the vein of other prior research that also focuses on the critical dynamics of the influence of business and labor money in American politics, such as Bauer, Pool, and Dexter 1963; Sorauf 1988; Box-Steffensmeier et al. 2005; Hertel-Fernandez 2014, 2018, 2019; Drutman 2015; Frymer and Grumbach 2020, we now include language to note that our processed dataset contains two types of PACs: business and labor (noted now when we introduce our data on page 5 in our marked-up revised manuscript, and also in the caption of Figure S1 presenting our dataset schema). In our processed dataset, there is a higher number of business PAC, and these PACs also donate to many more legislators on average than labor PACs. This is now clarified in the manuscript in the newly added S14 Appendix. 

We also conduct new machine learning experiments with legislator attributes or variables that consider only one type of PAC at a time, and show the issue-attention captured when only one type of PAC is considered. We show that business PACs offer a significantly higher association with issue-attention than labor PACs, but including both types of PACs is still significantly better than just considering business PACs. 

All these details are comprehensively provided in the new S14 appendix, and the aforementioned comparison for predictive capacity of issue-attention is presented in Figure S22 in our revised manuscript. We explain the classification scheme present in the original, as well as our processed data in the caption of Figure S1 (dataset schema) in our revised manuscript. This new analysis is noted when discussing our data in our marked-up revised manuscript (page 5), especially in footnote 10 that also points to S14 Appendix and Figure S22. 

Reviewer 2 notes that it might be helpful to include other legislator control variables such as: years in Congress, party leadership, marginal district, etc., and that more broadly, while there is an important discussion about the possibility of reverse causality in the conclusion, there is an additional concern about how the estimates are vulnerable to other forms of endogeneity like omitted variable bias. They ask us how including variables such as those mentioned above might affect our estimates. 

We agree with the Reviewer about the possibility of other variables not considered in this study, and thank them for highlighting several possible factors to consider, notably seniority and district marginality. To address this point, we have both conducted new empirical examinations and revised the text of the manuscript to reflect the additional attention to this issue. 

Since our framework allows for modeling each congressional session separately, we are able to consider several additional legislator attributes as mentioned by the Reviewer. In new work, we consider both the number of years a legislator has served in Congress (seniority) and a legislator’s district’s marginality (based on how their district votes in US presidential elections), and calculate these variables for individual congressional sessions. We do not find evidence that these attributes are associated with issue-attention (in terms of their predictive capacity on the held-out set), and our central finding that PACs offer the highest association with issue-attention continues to hold in these new specifications. Details on the construction of these new variables are provided in S15 Appendix, and the results for seniority and marginal districts are presented in Figures S23 and S24, respectively, in our revised manuscript. We also include pointers to these new additions in the Results section (page 11 of the marked-up version of our revised manuscript). 

In addition to the above revisions, in the Discussion section, we acknowledge the possibility that still other explanatory variables not included in this study could potentially play a role in predicting legislators’ issue-attention (page 14 in the marked-up version of our revised manuscript). However, since donor information offers significantly more predictive power than standard explanations, does so consistently over time, and constitutes a strong enough pattern that the automatic discovery of meaningful issue-PAC associations is possible, our results present an important finding and a foundation on which future work can build. In the manuscript, we suggest that future studies continue this research by considering other variables including attributes such as legislators' race and gender, or additional attributes of their districts. 

Additional edits.

We also addressed minor writing and editing errors, and appreciate Reviewer 1's close reading of the manuscript. We introduce the metadata once as examples in the introduction so as to avoid repetition on pages 4-5 in our revised manuscript (marked-up version). Removing this redundancy improves the writing. We also ensured that the link to Navient’s Wikipedia page (page 5 in original manuscript) works and directs to the correct page. We noticed that the text in the Wikipedia article has been changed or updated, so the first line borrowed from the Wikipedia page that we used in our submitted manuscript is no longer consistent with the Wikipedia page as it exists now. Therefore, we updated the sentence used in our revised manuscript to be consistent with the Wikipedia page we have linked (page 6 in the marked-up version). 

Finally, separate from the reviews, we note that several adjustments needed to be made to satisfy additional journal requirements as follows:

We confirm we have rechecked that our manuscript uses the official PLOS ONE LaTeX template and style requirements, and that the format of the names and affiliations complies with the style requirements as well.

We have no changes to make to the information provided in our Funding Statement. We have now removed funding information from the acknowledgements section of the revised manuscript (the removal is clearly marked in the marked-up copy of our manuscript that highlights changes from the original). There is no funding information in the text of the revised manuscript; it is provided in the Funding Statement alone as instructed.

Figures 4, S14 and S15 have been removed from the manuscript per the copyright issue, and have instead been replaced with Tables 1, S8 and S9, conveying the information we aimed to present in those figures using relevant text from properly provided sources. This should avoid any copyright issue as we are no longer using any images in these cases. 

Thank you to the editor and all the reviewers for their time and consideration.

Sincerely,

Pranav Goel, Nikolay Malkin, SoRelle W. Gaynor, Nebojsa Jojic, Kristina Miler, Philip S. Resnik

---

## [Decision Letter · Decision Letter 1]

17 Aug 2023

PONE-D-23-06208R1Donor activity is associated with US legislators’ attention to political issues

PLOS ONE

Dear Dr. Goel,

Thank you for submitting your manuscript to PLOS ONE. I have been assigned as the new editor to handle your manuscript. After careful consideration, I see that you have successfully revised the manuscript and that it is almost ready to be accepted. However, I would ask you to consider a minor change: You are very careful to talk about identified associations and not effects between PCA donations and speeches. Please abstain from causal language in point a) and b) at the beginning of your discussion. I know that you are not asking these questions as your research questions and rather questions one may generally wonder about, but putting them like this at the beginning of your discussion might invoke the idea that these are the questions you are answering (and thereby provide evidence for causality). I invite you to submit a revised version of the manuscript that addresses this point. Given that this is a very minor change, I hope you can update a revised version very quickly but latest by Oct 01 2023 11:59PM. There is no need to provide a response letter or track changes. Below, the standard letter continues. If applicable, we recommend that you deposit your laboratory protocols in protocols.io to enhance the reproducibility of your results. Protocols.io assigns your protocol its own identifier (DOI) so that it can be cited independently in the future. For instructions see: https://journals.plos.org/plosone/s/submission-guidelines#loc-laboratory-protocols. Additionally, PLOS ONE offers an option for publishing peer-reviewed Lab Protocol articles, which describe protocols hosted on protocols.io. Read more information on sharing protocols at https://plos.org/protocols?utm_medium=editorial-email&utm_source=authorletters&utm_campaign=protocols.

We look forward to receiving your revised manuscript.

Kind regards,

Mike Farjam

Academic Editor

PLOS ONE

Journal Requirements:

Reviewers' comments:

Reviewer's Responses to Questions

**Comments to the Author**

1. If the authors have adequately addressed your comments raised in a previous round of review and you feel that this manuscript is now acceptable for publication, you may indicate that here to bypass the “Comments to the Author” section, enter your conflict of interest statement in the “Confidential to Editor” section, and submit your "Accept" recommendation.

Reviewer #1: All comments have been addressed

2. Is the manuscript technically sound, and do the data support the conclusions?

Reviewer #1: Yes

3. Has the statistical analysis been performed appropriately and rigorously? 

Reviewer #1: Yes

4. Have the authors made all data underlying the findings in their manuscript fully available?

Reviewer #1: Yes

5. Is the manuscript presented in an intelligible fashion and written in standard English?

Reviewer #1: Yes

6. Review Comments to the Author

Reviewer #1: The authors should be commended for their rigorous revision. All of my previous reservations about the manuscript have been addressed through the new robustness checks. There is also reasonable support for the decision to use LDA modeling to conduct the particular type of study that they carry out in the methods section currently. Well done.

7. PLOS authors have the option to publish the peer review history of their article (what does this mean?). If published, this will include your full peer review and any attached files.

Reviewer #1: No

---

## [Author Response · Author response to Decision Letter 1]

17 Aug 2023

We have made the minor revision as requested, modifying the first paragraph of our discussion section to remove causal language so as not to mislead any reader. We appreciate the feedback from the reviewers and our editor.

---

## [Editor Report · Decision Letter 2]

24 Aug 2023

Donor activity is associated with US legislators’ attention to political issues

PONE-D-23-06208R2

Dear Dr. Goel,

We’re pleased to inform you that your manuscript has been judged scientifically suitable for publication and will be formally accepted for publication once it meets all outstanding technical requirements.

Kind regards,

Mike Farjam

Academic Editor

PLOS ONE

---

## [Editor Report · Acceptance letter]

29 Aug 2023

PONE-D-23-06208R2 

Donor activity is associated with US legislators’ attention to political issues  

Dear Dr. Goel:

I'm pleased to inform you that your manuscript has been deemed suitable for publication in PLOS ONE. Congratulations! Your manuscript is now with our production department. 

Kind regards, 

on behalf of

Dr. Mike Farjam 

Academic Editor

PLOS ONE